# Combination of Transcriptomics and Proteomics Reveals Differentially Expressed Genes and Proteins in the Skin of EDAR Gene-Targeted and Wildtype Cashmere Goats

**DOI:** 10.3390/ani13091452

**Published:** 2023-04-24

**Authors:** Gui-Zhen Gao, Fei Hao, Lei Zhu, Guo-Qing Jiang, Wei Yan, Jie Liu, Dong-Jun Liu

**Affiliations:** State Key Laboratory of Reproductive Regulation and Breeding of Grassland Livestock, School of Life Sciences, Inner Mongolia University, Hohhot 010070, China

**Keywords:** EDAR, cashmere goats, hair follicle, transcriptome, proteomics, regulatory network

## Abstract

**Simple Summary:**

The hair follicles of Cashmere goats can be divided into primary and secondary types owing to their occurrence time and structural characteristics. They show periodic growth, which includes three stages: anagen, catagen, and telogen. In the research on animal hair follicle growth and development, Cashmere goats serve as an important animal model. The EDAR gene, which targets Cashmere goats, has the phenotypic characteristics of abnormal hair growth and development at the top of the head. In this study, 732 and 140 differentially expressed genes and proteins, respectively, were identified using transcriptomic and proteomic techniques, providing important baseline data for understanding the mechanism of EDAR gene regulation in hair follicle growth and development.

**Abstract:**

Cashmere goats play a pivotal role in the animal hair industry and are economically valuable. Cashmere is produced through the periodic growth of secondary hair follicles. To improve their yield of cashmere, the regulatory mechanisms of cashmere follicle growth and development need to be analysed. Therefore, in this study, EDAR gene-targeted cashmere goats were used as an animal model to observe the phenotypic characteristics of abnormal hair growth and development at the top of the head. Transcriptomic and proteomic techniques were used to screen for differentially expressed genes and proteins. In total, 732 differentially expressed genes were identified, including 395 upregulated and 337 downregulated genes. In addition, 140 differentially expressed proteins were identified, including 69 upregulated and 71 downregulated proteins. These results provide a research target for elucidating the mechanism through which EDAR regulates hair follicle growth in cashmere goats. It also enriches the available data on the regulatory network involved in hair follicle growth.

## 1. Introduction

Cashmere goats are dual-purpose livestock that produce both cashmere and meat, thereby occupying an important position in animal husbandry [1]. Cashmere goats produce high-quality, white, fine, and soft cashmere, which has high economic value and effectively increases the income of breeders, hence promoting the development of the cashmere goat industry [2,3]. To maintain a high economic position, improving the yield and quality of cashmere is crucial since this is an important issue facing livestock production [4].

Hair follicles are accessory structures of animal skin [5]. Growth and shedding of animal hair are mediated by the periodic growth of hair follicles [6,7,8]. Hair follicles in cashmere goats can be divided into primary and secondary follicles based on the occurrence period and structural characteristics [9,10]. Wool and cashmere are formed by the growth and development of primary and secondary hair follicles, respectively [11,12]. Therefore, researchers can effectively improve cashmere production by altering the ratio of the secondary to primary hair follicles.

The occurrence and periodic growth of hair follicles are jointly regulated by multiple signalling pathways, including Wnt (Wnt/β-catenin), ectodysplasin-A receptor (EDAR), bone morphogenetic protein (BMP), fibroblast growth factor (FGF), Notch, and sonic hedgehog (Shh) [13,14,15]. Among these, EDAR plays an important role in regulating the development of animal ectoderm-derived organs, especially teeth, hair follicles, mammary glands, and sweat glands. Mutations of the EDAR gene can lead to hypohidrotic ectodermal dysplasia, which is characterised by sparse hair, lack of sweat glands, and tooth dysplasia [16]. The EDAR signalling pathway is mainly composed of the ligand ectodysplasin A (EDA), transmembrane receptor EDAR, and cytoplasmic adapter protein EDAR-associated adapter protein (EDARADD). EDAR and its specific ligand EDA belong to the tumour necrosis factor (TNF) receptor family and ligand family, respectively. They have a typical intracellular death domain of the TNF receptor family and can specifically bind to the death domain of the corresponding adapter protein EDARADD, initiating signal transduction and regulating downstream target gene transcription [17]. Previous studies in mice revealed that the EDAR gene is essential for the growth and development of primary hair follicles, with no effect on secondary hair follicles [18,19,20,21]. Whether a similar regulatory mechanism exists in the growth and development of hair follicles in cashmere goats warrants further investigation. If such a mechanism does exist, the ultimate result would be an increase in the rate of secondary hair follicles, which is expected to increase cashmere production.

In a previous study, EDAR gene-targeted cashmere goats were successfully created using CRISPR-Cas9 gene editing and somatic cell nuclear transfer technologies. The EDAR gene-targeted cashmere goats were observed to have the following typical characteristics: no hair on the top of the head, dry skin, and abnormal hair follicle development [22]. This also demonstrates that mutations in the EDAR gene have an important impact on the occurrence and development of hair follicles in cashmere goats. The EDAR gene-targeted cashmere goats can be used as an important animal model for studying hair follicle genesis and development, establishing a crucial foundation for future research on the underlying mechanism of hair follicle growth in cashmere goats.

In this study, EDAR-targeted cashmere goat skin samples were used as the experimental group, and wildtype cashmere goat skin samples were used as the control group. This study aimed to screen and analyse differentially expressed genes (DEGs) and proteins (DEPs) using transcriptomic and proteomic techniques, and to elucidate the mechanism through which EDAR regulates hair follicle growth and development in cashmere goats, thereby providing baseline data for further improving cashmere yield and quality.

## 2. Materials and Methods

### 2.1. Animal Ethics

The cashmere goats used in this study were raised at Yiwei White Cashmere Goat Co., Ltd. in Etuoke Banner, Ordos City, Inner Mongolia Autonomous Region, China. All the experimental procedures complied with the animal ethics regulations and guidelines of China. The experimental protocol was approved by the Animal Ethics Committee of the Inner Mongolia University.

### 2.2. Collection of Skin Samples from Cashmere Goats

The EDAR gene-targeted cashmere goats served as the experimental group, and the somatic cell nuclear transfer cashmere goats from the same cell line served as the control group. The selected animals were all male, 6-month-old Albas Cashmere goat breeds, raised under the same conditions. During the growth period of cashmere hair follicles (September), one goat from both groups was selected for anaesthesia. Skin samples were collected through surgical methods, with three duplicate samples per group. Skin samples from the head, back, and lateral sides of the experimental group goats were labelled EDAR01, EDAR02, and EDAR03, respectively. Skin samples from the same areas in the control group goats were labelled WT01, WT02, and WT03, respectively. All skin tissue samples were frozen in liquid nitrogen and transported to the laboratory for subsequent analysis.

### 2.3. RNA Sequencing (RNA-seq) Analysis

Total RNA was isolated from the skin tissue samples using TRIzol reagent (Invitrogen, Carlsbad, CA). RNA samples were validated using an Agilent 2100 RNA Nano 6000 assay kit (Agilent Technologies, Santa Clara, CA, USA). Validated RNA samples were enriched with mRNA, using magnetic beads with oligo (dT). Fragmentation buffer was added to the mRNA to make short fragments. The first-strand cDNA was synthesised using random primers and fragmented mRNA as a template. The second-strand cDNA was synthesised by adding dNTPs, RNase H, and DNA polymerase I. The cDNA was purified using a QIAQuick PCR kit (QIAGEN, Hilden, Germany) and eluted with EB buffer. The double-stranded cDNA was subjected to terminal repair, base-A addition, and sequence splicing. Target-size fragments were recovered via agarose gel electrophoresis and amplified via polymerase chain reaction (PCR). Finally, the entire library was prepared and sequenced on the Illumina platform (Illumina, San Diego, CA, USA).

Raw reads were obtained through sequencing on the Illumina platform. After removing low-quality sequences and connector contamination, clean reads were obtained. These sequences were compared with the reference genome using TopHat software (version 2.0.12, TopHat Software Pvt. Ltd., Indore, India) to locate them in the genome. The number of gene fragments in each sample was calculated using HTSeq (v0.6.0; Free Software Foundation, Sydney, Australia). The fragments per kilobase of exon model per million mapped fragments (FPKM) value was calculated to estimate the gene expression level in each sample.

DEGseq R package (v1.18.0; TNLIST, Beijing, China) was used for the analysis of DEGs between the samples. A *p*-value was assigned to each gene and adjusted using Benjamini and Hochberg’s approach to control for the false discovery rate. Genes with q ≤ 0.05 and |log2_ratio|≥ 1 were identified as DEGs [23]. Subsequently, the differential expression gene clustering Gene Ontology (GO) enrichment and Kyoto Encyclopedia of Genes and Genomes (KEGG) pathway analyses were conducted. The adjusted *p* < 0.05 for the GO terms and KEGG pathway was considered statistically significant [24,25].

### 2.4. Proteomic Analysis

The cashmere goat skin samples were ground into a cellular powder in liquid nitrogen, mixed with a lysis buffer (8 M urea, 1% protease inhibitor cocktail), and subjected to three ultrasound treatments on ice, using a high-intensity ultrasound processor. Debris was removed by centrifuging at 4 °C at 12,000× *g* for 10 min. Finally, the supernatant was collected, and the protein concentration was measured using a bicinchoninic acid kit, according to the manufacturer’s instructions.

The prepared protein solution required further trypsin digestion, which was carried out as follows: dithiothreitol was added to the protein solution to a final concentration of 5 mM and then reduced at 56 °C for 30 min. Iodoacetamide was added to achieve a final concentration of 11 mM, and the mixture was incubated at 25 °C in the dark for 15 min. Finally, the urea concentration of the sample was diluted to below 2 M. Pancreatin was added at a mass ratio of 1:50 (pancreatin:protein) and hydrolysed overnight at 37 °C. Trypsin was added at a mass ratio of 1:100 (trypsin:protein), and enzymatic hydrolysis was continued for 4 h [26].

The protein solution was digested with trypsin to form peptides. The peptides were reconstituted in 0.5 M triethylammonium bicarbonate and processed according to the manufacturer’s protocol for the tandem mass tag (TMT) kit/iTRAQ kit. Peptide segments were fractionated using high pH-based reverse high-performance liquid chromatography with Agilent 300 Extend C18 (5 μM particle diameter, 4.6 mm inner diameter, and 250 mm length). The procedure was as follows: the peptide segment grading gradient was 8–32% acetonitrile (pH 9), and 60 components were separated within 60 min. Subsequently, peptide segments were combined into nine components. The combined components were subjected to vacuum freeze-drying before subsequent analyses.

The peptide segments were separated using an ultra-high-performance liquid-phase system and injected into a capillary ion source for ionisation, followed by analysis using timsTOF Pro mass spectrometer. The ion source voltage was set at 1.4 kV, and the peptide parent ions and their secondary fragments were detected and analysed using TOF. The scanning range of the secondary mass spectrometer was set at 100–1700 m/z. Parallel cumulative serial fragmentation (PASEF) mode was used for data acquisition. After collecting the primary mass spectrum, the PASEF mode was repeated 10 times to collect a secondary spectrum with the charge number of the parent ion in the range of 0–5. The dynamic exclusion time for tandem mass spectrometry scanning was set to 24 s to avoid repeated scanning of parent ions [27].

Secondary mass spectrometry data were retrieved using Maxquant (v1.6.5.0; Max-Planck-Institute of Biochemistry, Munich, Germany). Retrieval parameter settings were as follows: UniPort Capra_ hircus (35,264 sequences) added a reverse library to calculate the false discovery rate caused by random matching, and it also added a common contamination library to the database to eliminate the impact of contaminated proteins on the identification results. The enzyme digestion method was set to Trypsin/P, the number of missing bits was set to 2, the mass error tolerance of primary parent ions for the first search and the main search was set to 70 ppm for both, and mass error tolerance of secondary fragment ions was 0.04 Da. The alkylation of cysteine was set as a fixed modification, and the variable modifications were the oxidation of methionine and acetylation of the N-terminus of the protein. The false discovery rate for protein and peptide-spectrum matching identification was set to 1%. DEPs were annotated using the UniProt-GOA database http://www.ebi.ac.uk/GOA/ (accessed on 20 May 2019). GO and KEGG enrichment analyses were performed simultaneously.

### 2.5. Quantitative Real-Time PCR (qRT-PCR)

Total RNA from each skin sample was prepared using the RNAiso Plus reagent (Takara Bio Inc., Shiga, Japan; Code no. 9108). Then, the genomic DNA was removed, and the cDNA templates were prepared according to the instructions of the reverse transcription PrimeScript RT reagent kit with gDNA Eraser (Takara; Code no. RR047A). The experiment was performed according to the instructions of the real-time quantitative PCR detection kit, TB Green Premium Ex Taq II (Takara; Code no. RR420A). The genes detected in the experiments were DEGs obtained through transcriptomic analysis. The primers used to detect each gene are listed in Appendix A. The glyceraldehyde 3-phosphate dehydrogenase (GAPDH) gene served as an internal reference gene, and the 2^−△△Ct^ method was used to calculate the relative expression of each gene [28].

### 2.6. Haematoxylin–Eosin (H&E) Staining

Skin tissue samples from cashmere goats were fixed with 4% paraformaldehyde for 24 h. Then, they were treated with a gradient concentration of ethanol for dehydration, subjected to xylene treatment, and completely embedded in paraffin. Embedded tissue samples were sectioned at a thickness of 5 µm using a paraffin-embedded sectioning machine. The prepared paraffin sections were sequentially treated with haematoxylin, 1% hydrochloric acid, 1% ammonia, and 0.5% alcohol-soluble eosin. Lastly, the sections were sealed with neutral gum and observed under a microscope. The transverse and longitudinal sections of the hair follicles on the cashmere goat skin samples could now be clearly observed [29].

### 2.7. Statistical Analyses

Each experiment was repeated thrice, and the relative mRNA expression of DEGs in different skin samples was statistically analysed using GraphPad Prism 8 (GraphPad 8.0.1 software, San Diego, CA, USA). Data were expressed as the mean ± standard deviation. Statistical significance was set at *p* < 0.05.

## 3. Results

### 3.1. Phenotypes of EDAR Gene-Targeted Cashmere Goats

After birth, EDAR gene-targeted cashmere goats had a unified phenotype, with dry skin and no hair on the top of their head. At six months, compared with the wildtype, the EDAR gene-targeted cashmere goats still had sparse hair on their heads and dry skin (Figure 1a). The head, back, and lateral skin samples of the EDAR gene-targeted and wildtype cashmere goats were collected, paraffin sections were prepared, and haematoxylin and eosin staining was performed. The results showed that primary and secondary hair follicles could be found in the head skin tissue of EDAR gene-targeted cashmere goats; however, compared to the wildtype, the number of hair follicles per unit skin area was significantly lower. Primary and secondary hair follicles were also found in the back and lateral skin of the EDAR gene-targeted cashmere goats, although compared to the wildtype, the morphological structure of the hair follicles was smaller (Figure 1b).

### 3.2. Analysis of Differences in Gene Expression

Using the Illumina sequencing platform, transcriptome sequencing analysis was performed on six skin samples from EDAR gene-targeted and wildtype cashmere goats. In total, 285,039,622 raw reads were obtained. In all the sequenced samples, the proportion of bases with a mass value greater than 30 (an error rate of less than 0.1%) was above 90% (Appendix A). After filtering the data, 277,261,506 clean reads were obtained. Compared to the reference genome of cashmere goats, 247,462,009 mapped reads were obtained, with a comparison rate of 89.25% (Table 1).

DEGs were determined by comparing gene expression in each sample. In the head skin of EDAR01 and WT01 samples, there were 4899 DEGs, of which 2441 were upregulated and 2458 were downregulated. In the back skin of EDAR02 and WT02 samples, there were 2151 DEGs, of which 1302 were upregulated, and 849 genes were downregulated. In the lateral skin of EDAR03 and WT03 samples, there were 2094 DEGs, of which 1117 were upregulated, and 977 were downregulated. Overall, 732 DEGs were found in various skin samples of EDAR gene-targeted and wildtype cashmere goats, of which 395 were upregulated, and 337 were downregulated (Figure 2a,c and Appendix A). Cluster analysis was conducted for DEGs among the samples. The results showed that the DEGs had different expression patterns in the head and other skin locations, which is consistent with the phenotype of EDAR gene-targeted cashmere goats (Figure 2b).

The mRNA expression levels of GHRHR, IL36B, JAKMIP1, JSRP1, MAP34-B, MOGAT1, PAEP, and TYRP1 in the skin samples of EDAR gene-targeted and wildtype cashmere goats were compared using qRT-PCR. The results showed a consistent trend of change compared to the transcriptome analysis results (Figure 3).

All DEGs passed GO functional analysis. The category of biological processes were mainly distributed into three modules: cellular processes, single-organism processes, and biological regulation. In the category of cell components, they were mainly distributed in three modules: the cell, organelle, and membrane. Lastly, in the category of molecular function, they were mainly distributed in three modules: binding, catalytic, and transporter modules (Figure 4 and Appendix A).

### 3.3. Analysis of Differences in Protein Expression

To reveal the cause of phenotypic formation in EDAR gene-targeted cashmere goats, TMT combined with liquid chromatography–tandem mass spectrometry (LC–MS/MS) was used to detect DEPs in all goat skin samples. In total, 107,499 secondary spectra were obtained using mass spectrometry. After searching the protein theoretical data library, the available effective spectrum number of the secondary spectrum of mass spectrometry was 28,648, and the spectrum utilisation rate was 26.6%. A total of 21,909 peptide segments were identified via spectral analysis, of which the specific peptide segments were 20,166. A total of 3750 proteins were identified, of which 2912 contained quantitative information (Figure 5a).

DEPs were identified by quantitative mass spectrometry-based protein analysis. In the EDAR01/WT01 group, 177 proteins were upregulated, and 176 were downregulated. In the EDAR02/WT02 group, 123 proteins were upregulated, and 129 were downregulated. In the EDAR03/WT03 group, 251 proteins were upregulated, and 260 were downregulated. In the EDAR/WT group, 69 proteins were upregulated, and 71 were downregulated (Figure 5b,c, Appendix A).

The subcellular localisation of DEPs was predicted using wolfpsort. A total of 44 proteins (31.43%) were located in the cytoplasm, 44 (31.43%) in the extracellular region, 14 (10%) in the plasma membrane, 12 (8.57%) in the nucleus, and 11 (7.86%) in the mitochondria (Figure 6a).

The distribution of DEPs in the GO secondary annotation was statistically analysed, where a *p*-value of less than 0.05 was considered significant. In the category of biological processes, DEPs were mainly distributed in three modules: metabolic processes, single-organism processes, and cellular processes. In the category of cell components, DEPs were mainly distributed in three modules: the cell, membrane, and extracellular region. In the category of molecular function, DEPs were mainly distributed in three modules: binding, catalytic activity, and transporter activity (Figure 6b and Appendix A).

### 3.4. Enrichment Analysis of DEGs and DEPs in KEGG

In the KEGG pathway, DEGs were mainly enriched in drug metabolism, cytochrome P450, neuroactive ligand–receptor interaction, retinol metabolism, complexion and coagulation cascades, nicotine addiction, haematopoietic cell lineage, tyrosine metabolism, and metabolism of xenobiotics, via cytochrome P450 signalling pathways. The DEPs were mainly enriched in the signalling pathways of drug metabolism, cytochrome P450, cystine, and thionine metabolism; drug metabolism, including other enzymes, arginine and proline metabolism, tyrosine metabolism, glycine, serine, and threonine metabolism, and vitamin B6 metabolism. In particular, there was an intersection between the DEGs and DEPs in the KEGG pathway at the drug metabolism cytochrome P450 signalling pathway (Figure 7).

## 4. Discussion

We observed the phenotypic characteristics of EDAR gene-targeted and wildtype cashmere goats and compared the structural characteristics of their skin follicles. The results revealed significant phenotypic differences between them, especially in the head skin samples. At the age of 6 months, EDAR gene-targeted cashmere goats had hair on their heads, although the number of hair follicles was low, and the structure of hair follicles on the back and sides of their bodies was small, in comparison to the wildtype cashmere goats. It can be inferred that the deletion of the EDAR gene may have an impact on the early development of hair follicles in cashmere goats, although as individuals continue to develop, a regulatory network compensates for this, slowing hair follicle development [30].

In this study, we obtained transcriptomic and proteomic data from the goat skin samples and screened the DEGs and DEPs. Through GO functional enrichment analysis, we found that in the biological process category, the DEGs and DEPs were mainly enriched in the cellular process and single organism process modules; in the category of cell components, they were mainly enriched in the cell and membrane modules. In the molecular function category, they were mainly enriched in the binding and catalytic modules. Therefore, it was evidenced that the deletion of the EDAR gene significantly changed the expression of genes closely related to the above terms. Simultaneously, there was significant enrichment of DEGs and DEPs in neuroactive ligand–receptor interaction, retinol metabolism, drug metabolism, and cytochrome P450 signalling pathways.

Neuroactive ligand–receptor interaction signalling pathways include all ligand receptors involved in the intracellular and extracellular signalling pathways in the plasma membrane of a cell [31,32]. In EDAR gene-targeted cashmere goat skin samples, DEGs related to the signalling pathway of neuroactive ligand–receptor interactions were both upregulated and downregulated. For example, the expression of the calcitonin receptor was upregulated, whereas the expression of the growth hormone-releasing hormone receptor was downregulated, which is consistent with the significant enrichment of DEGs in molecular function in GO analysis. These signalling pathways play an important role in the formation and maintenance of tissues and organs. For example, studies have shown that mice transfected with insulin-like growth factor 1 have improved hair growth [33,34]. The occurrence and development of hair follicles are affected by multiple signalling pathways. In this study, we found abnormal expression of ligand receptor genes in several different membranes. Therefore, it can be deduced that the deletion of the EDAR gene led to changes in the expression of ligand receptors in multiple signalling networks, leading to abnormalities in hair follicle development.

The retinol metabolism pathway is the main pathway for vitamin A metabolism [35,36]. Vitamin A not only maintains normal visual function but also has a close relationship with the differentiation of epithelial cells. Studies have shown that vitamin A promotes hair follicle growth by stimulating hair follicle stem cells to induce the occurrence and prolongation of hair growth during the hair cycle. Therefore, in the skin samples of EDAR gene-targeted cashmere goats, DEGs were enriched in this signalling pathway, indicating that abnormalities in hair follicles may be related to changes in the expression of genes related to retinol and vitamin A metabolism.

The cytochrome P450 signalling pathway is very specific to drug metabolism. Both DEGs and DEPs were enriched in this signalling pathway. The expression of genes related to citalopram metabolism was upregulated. Some studies have found that citalopram can cause hair loss when used to treat depressive mental disorders, indicating that citalopram metabolites affect hair growth [37]. This was evidenced by the hair growth abnormalities that occurred in EDAR gene-targeted cashmere goats.

In summary, the occurrence and development of hair follicles are regulated by a series of signalling networks. Changes in the expression of key genes trigger changes in the expression of related genes in the regulatory network, ultimately leading to phenotype generation. Our findings revealed a phenotype of abnormal hair follicle development in EDAR gene-targeted cashmere goats, which was the result of a series of related gene expression changes caused by the deletion of the EDAR gene. Such analyses of DEGs and DEPs in the skin provide powerful reference data for further elucidating the mechanism of hair follicle occurrence and development.

## 5. Conclusions

In this study, transcriptomic and proteomic techniques were used to compare DEGs and DEPs in the skin samples of EDAR gene-targeted and wildtype cashmere goats. We identified 732 DEGs, including 395 upregulated and 337 downregulated genes. In addition, 140 DEPs were identified, of which 69 were upregulated, and 71 were downregulated. Our study provides a target for elucidating the mechanism through which EDAR regulates hair follicle growth in cashmere goats. It also enriches our understanding of the regulatory network involved in hair follicle growth.

## Figures and Tables

**Figure 1 animals-13-01452-f001:**
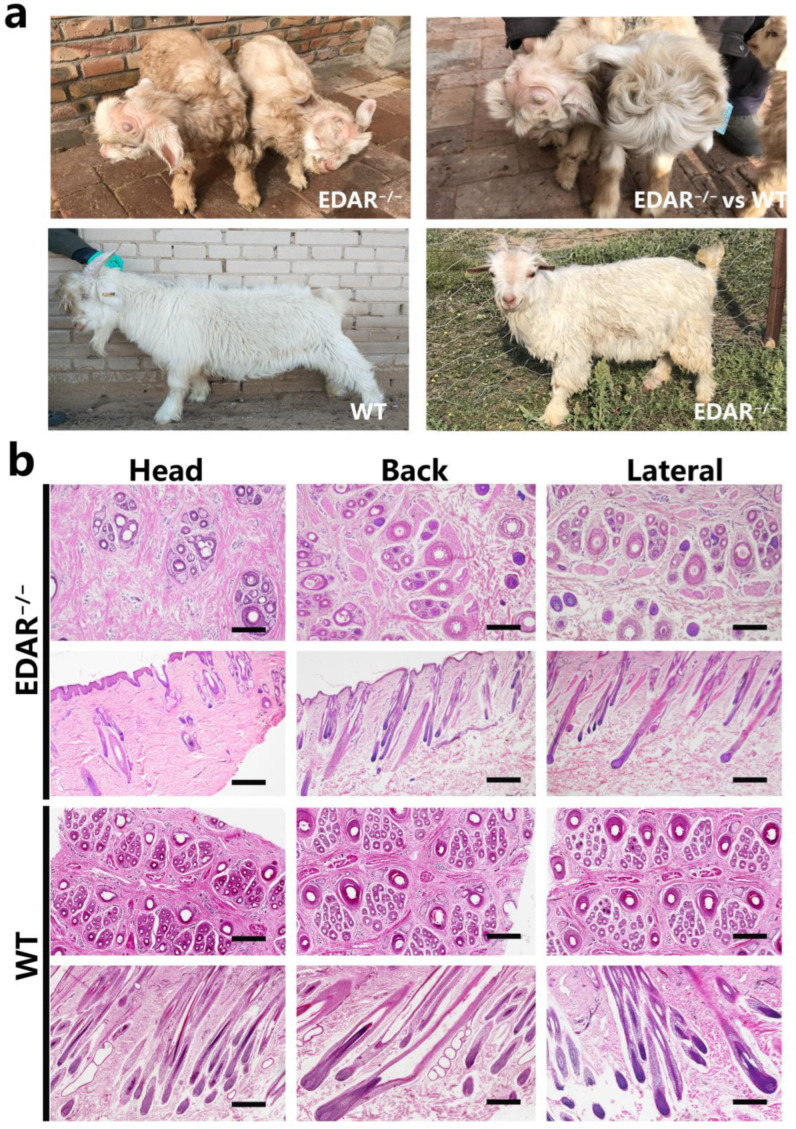
EDAR gene-targeted cashmere goats and skin tissue sections. (**a**) EDAR gene-targeted cashmere goats. The top left photo shows two newly born goats, which displayed the characteristic hairlessness on their heads. The top right panel shows a comparison between newly born EDAR gene-targeted and wildtype cashmere goats. The lower left image shows a 6-month-old wildtype cashmere goat and the lower right image shows a 6-month-old EDAR gene-targeted cashmere goat; (**b**) Cross-sectional and longitudinal-sectional images of cutaneous tissues from different body parts of the EDAR gene-targeted and wildtype cashmere goats. Scale bar = 100 μm.

**Figure 2 animals-13-01452-f002:**
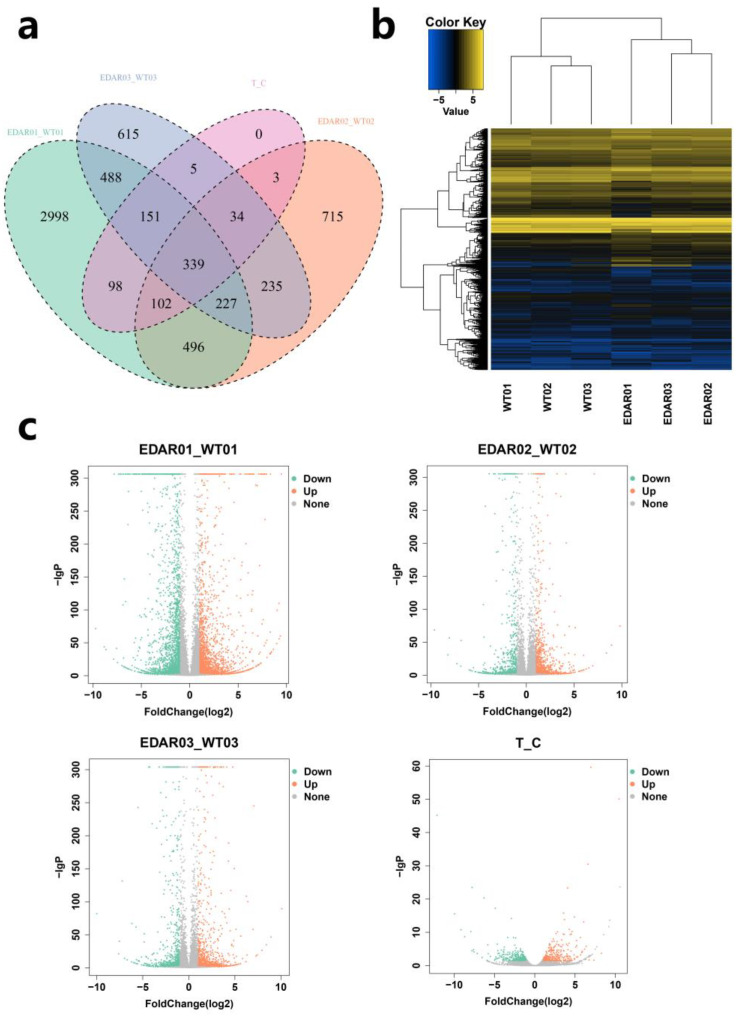
Identification of differentially expressed genes (DEGs) in cashmere goat skin samples. (**a**) Venn diagram of DEGs between groups; (**b**) cluster diagram of DEGs between groups. (**c**) Volcano diagram of different genes between groups.

**Figure 3 animals-13-01452-f003:**
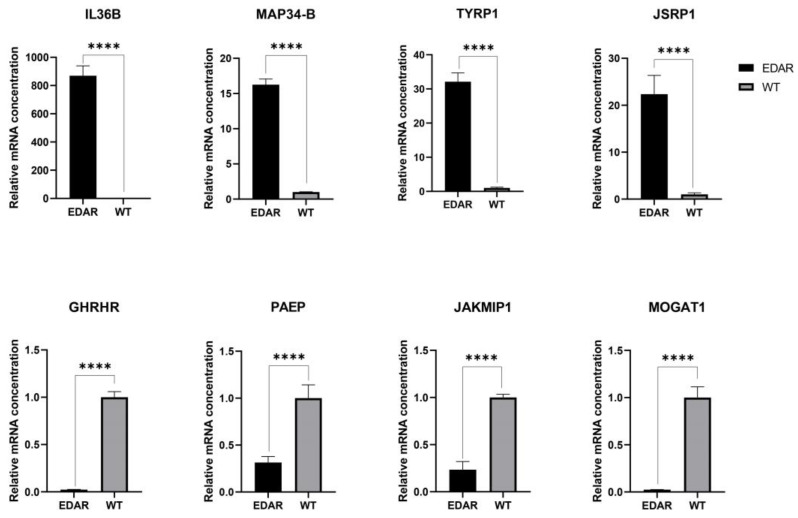
mRNA expression levels for eight genes in the skin samples of EDAR gene-targeted and wildtype cashmere goats examined, via quantitative polymerase chain reaction, to verify the RNA sequencing data. GAPDH served as the reference gene to normalize expression levels. Results are expressed as fold changes. Each bar represents the mean ± standard deviation (SD) of three samples. Significant differences are indicated by an asterisk (**** *p* < 0.001). GHRHR: growth hormone releasing hormone receptor; IL36B: interleukin 36 beta; JAKMIP1: janus kinase and microtubule interacting protein 1; JSRP1: junctional sarcoplasmic reticulum protein 1; MAP34-B: MAP34-B protein; MOGAT1: monoacylglycerol O-acyltransferase 1; PAEP: progestagen-associated endometrial protein; TYRP1: tyrosinase-related protein 1.

**Figure 4 animals-13-01452-f004:**
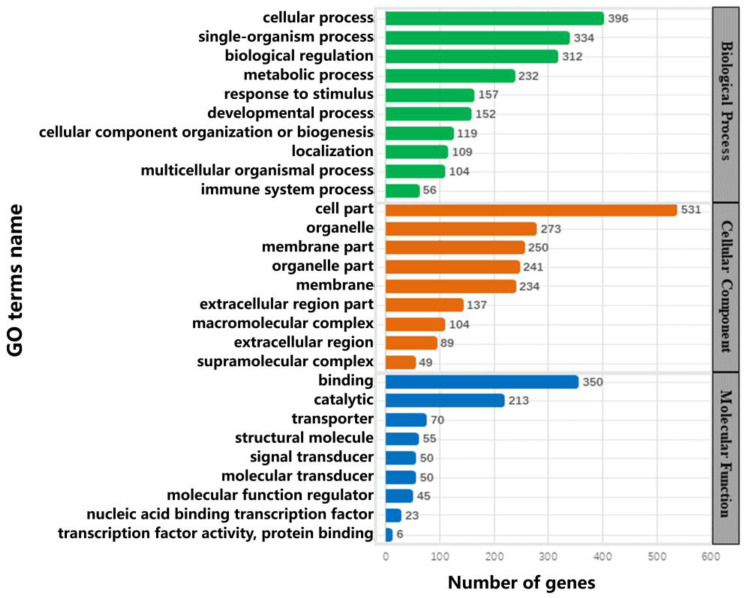
Enriched Gene Ontology (GO) terms for the differentially expressed genes.

**Figure 5 animals-13-01452-f005:**
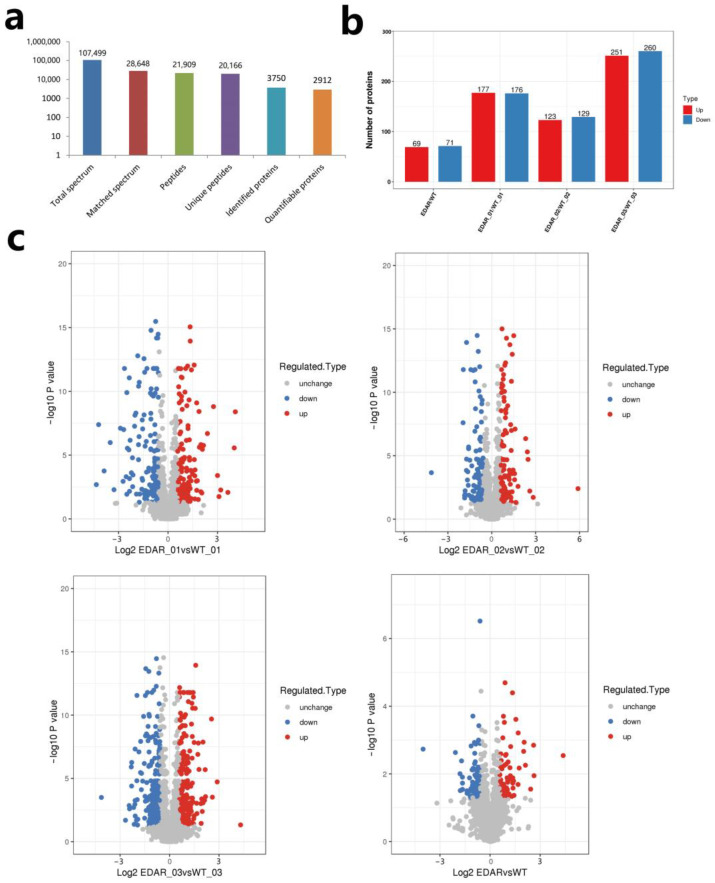
Identification of differentially expressed proteins (DEPs) in cashmere goat skin samples. (**a**) Mass spectrometry data statistics of skin proteins from EDAR gene-targeted and wildtype cashmere goats; (**b**) statistical information on DEPs in skin samples of EDAR gene-targeted and wildtype cashmere goats; (**c**) volcano diagram of different proteins between groups.

**Figure 6 animals-13-01452-f006:**
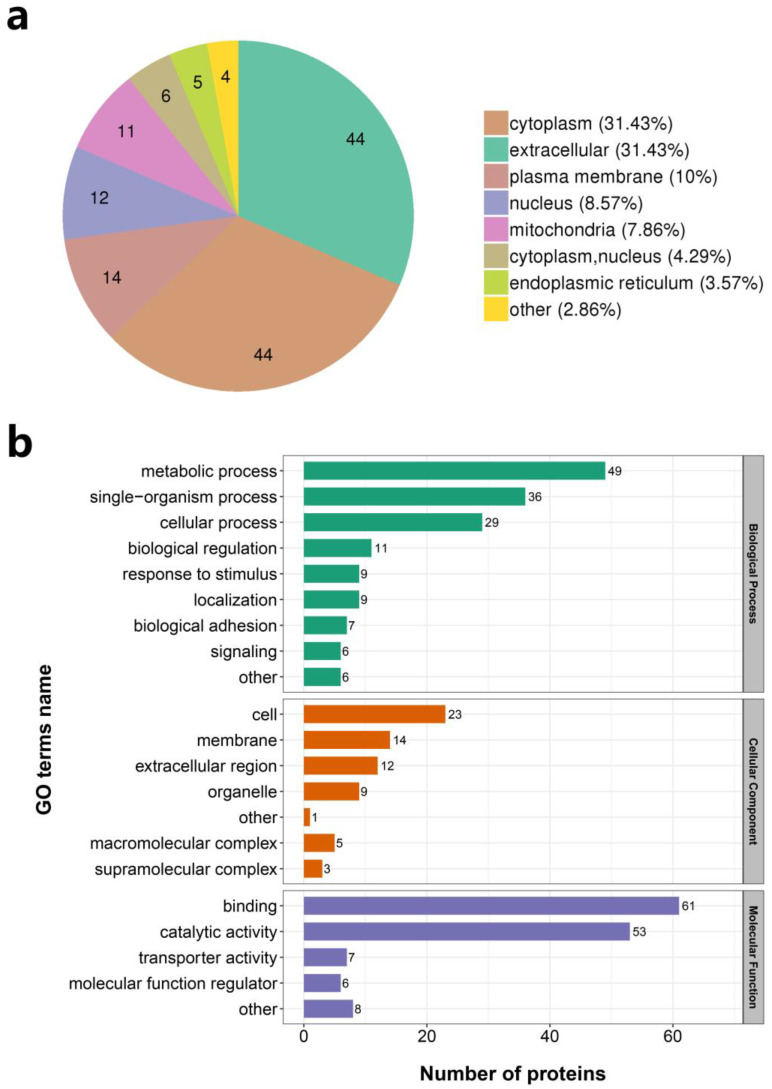
Enrichment analysis of differentially expressed proteins (DEPs). (**a**) Subcellular structural mapping of DEPs in skin samples of EDAR gene-targeted and wildtype cashmere goats. (**b**) Enriched Gene Ontology (GO) terms for the DEPs.

**Figure 7 animals-13-01452-f007:**
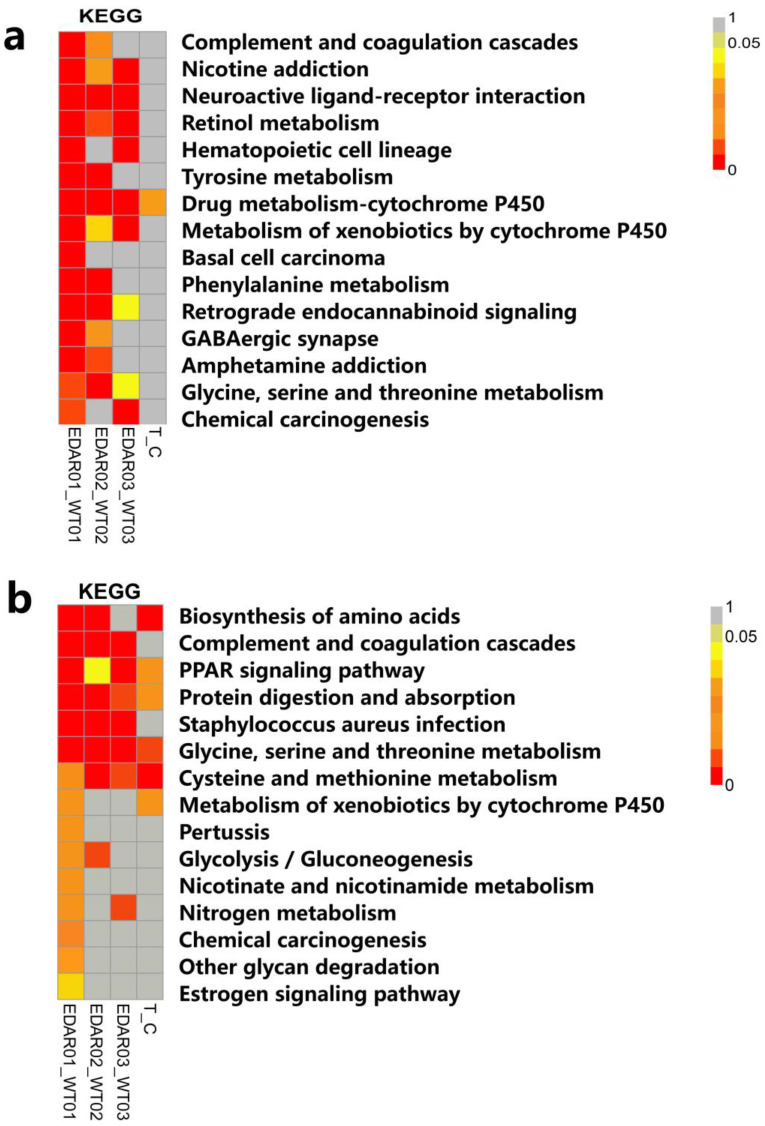
Enrichment analysis of differentially expressed genes (DEGs) and proteins (DEPs) in the Kyoto Encyclopedia of Genes and Genomes (KEGG). (**a**) Enrichment analysis of DEGs in the KEGG signalling pathway in the skin tissues of EDAR gene-targeted and wildtype cashmere goats; (**b**) enrichment analysis of DEPs in the KEGG signalling pathway in the skin tissues of EDAR gene-targeted and wildtype cashmere goats.

**Table 1 animals-13-01452-t001:** The mapping of clean data.

Library	Total Reads	Mapped Reads	Mapping Rate
EDAR01	47,835,374	42,692,760	0.8925
EDAR02	44,430,710	40,183,265	0.9044
EDAR03	46,326,914	41,132,985	0.8879
WT01	47,459,130	41,738,567	0.8795
WT02	45,578,664	40,875,932	0.8968
WT03	45,630,714	40,838,500	0.895

## Data Availability

All data are presented in the manuscript.

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
