# Peer review of "Combination of Transcriptomics and Proteomics Reveals Differentially Expressed Genes and Proteins in the Skin of EDAR Gene-Targeted and Wildtype Cashmere Goats"

_animals, 2023, doi:10.3390/ani13091452_

Round 1

Reviewer 1 Report

The manuscript by Gao etc., investigated differentially expressed genes/proteins in the skin of EDAR gene-edited goats. The experiments were straightforward and were carried out on unique materials. Nevertheless, the manuscript lacks some analytical details and needs revisions to improve clarity, it also needs a major English revision. Examples of these issues are given below to improve the manuscript.

1, Hair follicles exhibit periodic growth characteristics. The timing of skin sampling was not specified in the manuscript. Please also explain the specific reasons for the timing of sample collection.

2, Please advise the genotype before and after EDAR editing in the manuscript, as well as the directionality of upregulated and downregulated genes.

3, What is the criteria for selecting genes for qPCR validation?

4, Redundant information was provided in table2 and figure2b.

5, I would recommend adding enrichment statistics for figure4 and figure6, and use the same graphic design for figure7a and 7b.

The English needs to be improved.

Author Response

Dear reviewers:
Thank you for your letter and for your comments concerning our manuscript entitled “Combination of transcriptomics and proteomics reveals differentially expressed genes and proteins in the skin of cashmere goats targeted by EDAR genes” (Submission ID: animals-2353876). This comment is valuable and important guiding significance to our researches. According with your advice, we tried our best to amend the relevant part and made some changes in the manuscript. These changes will not influence the content and framework of the paper. All of your questions were answered in the attached materials below.  

Once again, thank you very much for your comments and suggestions.

Yours Sincerely,
Dongjun Liu

Reviewer 2 Report

The manuscript seems lack many important information to judge whether the content was valid. For example...

> There are EDER gene(s) in title and throuout the manuscript. However, these seems no detailed explanation about the EDER gene. What was this?

>The authors did not provide the detailed description  in Materials and Methods section about the samples used., including the number of samples and pedigree structure as well ass feeding conditions, age, and gender.

>The resolution of the figures should be improved.

> What was the novel finding(s) from this study? Prease clarify.

Minor english proofreading might be encouraged. For example, I found meaningless [;] in the legend of Figure 3 (P8L279).

Author Response

(The authors gave the same response as above.)

Round 2

Reviewer 2 Report

The manuscript has been improved.